# A Lexicon of Descriptive Sensory Terms for Peas (*Pisum sativum* L.): A Systematic Review

**DOI:** 10.3390/foods13142290

**Published:** 2024-07-20

**Authors:** Szymon Wojciech Lara, Amalia Tsiami

**Affiliations:** 1London Geller College of Hospitality and Tourism, University of West London, St. Mary’s Road, Ealing, London W5 5RF, UK; 2Royal Botanic Gardens, Kew, Richmond, London TW9 3AE, UK

**Keywords:** sensory cues of peas, *Pisum sativum* L., *Lathyrus oleraceus* Lam, descriptive sensory terms for peas, sensory properties of peas, peas sensory lexicon

## Abstract

Background: The popularity of peas (*Pisum sativum* L.) and pea-derived products is constantly growing globally and is estimated to continue to do so at an average annual rate of 12%. This is partially stimulated by the increase in the consumption of meat analogues and the popularisation of animal-protein-free diets. Peas are considered a great source of protein and dietary fibre and are not depicted as allergenic, making them a good replacement ingredient for other legumes such as soy. Peas are also considered good for the environment, mainly due to their nitrogen fixation capabilities. Despite the above benefits, sensory quality is still a limiting factor in increasing consumer acceptance of peas and pea-derived products. Results: This review has been conducted in accordance with the Joanna Brings Institute’s guidance for systematic literature reviews. The search has been conducted on the descriptive sensory terms for *Pisum sativum* L., where the objectives of the study were to select, present, and analyse the identified descriptive sensory terms for peas found throughout the academic literature. The reviewers have screened 827 articles, of which 12 were eligible for data extraction. Out of the 12 articles, 205 descriptive sensory terms were identified. Those were divided into five categories: smell/odour (27%), flavour (51%), taste (10%), texture (8%), and visual (4%). These included results from sensory analyses by trained/untrained panels and instrumental analyses of texture and of volatile compounds. Conclusion: The identified descriptive sensory terms for *Pisum sativum* L. could be used for future descriptive sensory evaluation of peas and other legumes, making the process less time consuming. The full list could be used for the initial sensory panel training and then adapted based on the frequency of the depicted terms that meet the criteria for the developed lexicon.

## 1. Introduction

The objective of this review is to screen and review existing terminologies in use for the descriptive sensory evaluation of peas and to summarise and form a pea-specific lexicon of descriptive terms.

Pulses belong to the leguminous family (Fabaceae), and the most commonly cultivated ones are *Cicer arietinum* (chickpeas), *Lens culinaris* L. (lentils), *Lupinus* spp. L. (lupins), *Vicia faba* L. (faba beans, broad beans), *Phaseolus vulgaris* L. (common beans like kidney bean etc), *Glycine max* L. (soybeans), *Arachis hypogea* L. (peanuts), and *Pisum sativum* L. (peas) [1,2]. Of note, *Pisum sativum* L. has also been recently reassigned taxonomically as *Lathyrus oleraceus* Lam [3]. However, this is not yet widely adopted. Peas were domesticated around 10,000 years ago in today’s Middle East/Middle Asia, Mesopotamia, and potentially the central plateau of Ethiopia [4,5,6]. Their cultivation then spread to other parts of the world, and today, hundreds if not thousands of cultivated varieties of *Pisum sativum* L. can be found around the globe [7].

Peas are mainly grown for fresh and dried seed production. and the current world production is estimated to be around 10 million tons and 7 million tons, respectively [8,9]. According to the Food and Agriculture Organisation [10], in 2022, China was the biggest pea-producing country, followed by India, USA, France, and Egypt. The pea market is expected to grow by around 5% between the years 2022 and 2027 [10,11]. Reports from markets producing pea by-products and derivatives state that the demand for peas is constantly accelerating [6,8,10,11]. This is likely to be caused by the growth of popularity and usage of products like pea protein isolates in meat and dairy analogues, as these markets are expected to grow at an annual average of 12% [11,12]. Pea compounds are being used in food reformulation and product development sectors, particularly to produce meat-analogues, i.e., vegan foods, which are often marketed as “healthier”, as “high in nutritional compounds”, and as “sustainable” foods [13]. 

The macronutrient composition of peas is often used as an indicator of the importance of peas in global food systems. The typical composition of most *Pisum sativum* L. cultivars is as follows: protein (21.2–32.9%), starch (36.9–49%), amylose (20.7–33.7%), dietary fibre (14–26%), soluble sugars (5.3–8.7%), and lipids (1.2% to 2.4%) [14]. The protein and fibre contents, when compared to other popular vegetables, are significantly higher, making peas an important source of these elements for human consumption, especially for those that avoid the consumption of animal-derived protein. Furthermore, peas are not required to be declared as allergenic in Europe, the USA, and some other parts of the world, making peas a less-risky investment (adverse-reactions to ingredients) for stakeholders, especially when compared to other allergen-labelled legumes like soy [15]. 

Apart from the above, legumes, and especially peas, have been publicised as the right crops for tackling future food and nutrition insecurity [16]. Through the agricultural lens, the aspect of nitrogen fixation capabilities that peas pose is exploited extensively, making them an ideal break crop for monocultures [17]. Legumes like peas support the growth of nitrifying bacteria (e.g., *Nitrosomonas* and *Nitrobacter*), which convert *ammonium ions* to *nitrate ions* and *oxides nitrite ions* to *nitrate ions*, resulting in feeding the soil humus, producing nutrients available to others, and more susceptible crops [18]. The pea crop is considered a “climate smart” crop, often withstanding periodic weather fluctuations, but its most optimal environment is the cool climate, with a minimum of 10 °C and a maximum of 23 °C. Nevertheless, numerous varieties of peas can withstand climatic conditions outside of these ranges, including biotic/abiotic stresses, for example, varieties from northeast and sub-Saharan Africa [10,18,19]. The rich bank of varieties of peas has the potential to be used as “filler crops” for current and future agriculture, simultaneously contributing to the global–local increase in agrobiodiversity [7,20,21].

The consumer acceptance element is regarded as one of the highest factors, as diminished consumer acceptance would decrease economic prosperity of the sector. Apart from testing the microbiological, chemical, allergenic, and physical properties of the products, businesses also evaluate their products’ descriptive sensorial acceptability [22,23]. 

The sensory characterisation of peas can be conducted either instrumentally, with the help of scientific instruments and methods such as gas chromatography (G-C), texture analysis, ash analysis, and others, or through the help of panel-based descriptive sensory profiling of products [24]. The smell and taste cues, when combined, are referred to as the flavour. The following are some non-flavour-based sensory characteristics that can influence the sensory perception of flavours: tingling, dry, rough, metallic, pungent, spicy, and astringent; these are either caused by the temperature, moisture, surface/texture, or the chemical composition of the product including the volatile compounds [24,25]. Sensory analysis of products, especially executed with a sensory panel, should be based on a pre-determined and agreed-on list of sensory ques. The descriptive lexicon can be developed by the panel, often a trained panel, or it can be derived from the literature. Volatile compounds are also extremely important as they are susceptible to significant changes due to the processing of peas, such as *cis-3-hexenal* or *1-Octen-3-ol*, which are both linked to disliked sensory characteristics in peas like “grassy”, “beany”, or “boiled” odours and many off-flavours released due to heat processing.

A standardisation of the process, specifically using the same list of sensory ques for similar products, when conducting comparison tests, is recommended. This review hopes to collect, revise, and form a descriptive sensory lexicon for peas specifically, not legumes in general, as such has not been found in our literature screening, making the process of pea products’ descriptive sensory profiling less difficult and more accurate in the future.

## 2. Methodology 

This review was conducted in accordance with the Joanna Briggs Institute’s guidance for systematic reviews for effective data synthesis [26]. The literature search was conducted electronically on the following scientific data bases: (1) Science Direct; (2) Emerald Insight; (3) ProQuest; (4) PubMed; and (5) Summon Search Engine (provided on behalf of University of West London). The searches were conducted across each database using 1 Boolean Code:*(pisum sativum L. sensor* analysis*) or (pea* sensor* analysis*) or (pea* sensor* cue*) or (pea* flavour*) or (pea* flavor*) or (pea* smell*) or (pea* odourant*) or (pea* taste*) or (pea* texture*) or (pea* visual character*) or (pea* colour*) NOT (pigeon pea) NOT (Cajanus cajan L.)*

The searches were limited to peer-reviewed and published journal articles only; grey literature was not considered for inclusion. The searches were not limited by any date restraints, the geographical location of the published articles, or the area of study. Articles were, however, required to be written in English. 

### 2.1. Shortlisting, Screening and Selection of Evidence

The search results were undertaken for a stage 1 shortlisting, where the titles and abstracts of the articles were screened. If the information within the title and abstract of the study seemed relevant to the review’s objectives, which were to identify, analyse, and present the applied descriptive sensory terms for peas, then the study was considered for stage 2 screening. Stage 2 screening was carried out using a software programme Zotero 6.0, where the shortlisted articles were subjected to a full-text screening. Articles considered as appropriate for inclusion were transferred into a software program NVivo 12 Pro, where appropriate fragments of text and information were highlighted, extracted, and analysed for the results and discussion chapters. The search, shortlisting, and selection processes have been presented in the PRISMA flow chart under Figure 1.

### 2.2. Inclusion Criteria

Inclusion criteria were used throughout the search, shortlisting, and selection of evidence phases. These consisted of the following: the selected articles had to include information on the applied descriptive sensory terms for peas (*Pisum sativum* L.), and existing descriptive sensory terms had to be developed through a panel-based descriptive sensory profiling of peas and derived products, unless derived through a literature review, and through appropriate instrumental analyses (please see Appendix A). 

Articles that were not based on peas but on other products, or those that did not specify the nature of the tested product, were excluded from the review. Articles that included information on descriptive sensory terms for peas but with no indication of sensory evaluation by panellists or instruments and were not classified as literature reviews were also removed from the search.

### 2.3. Data Presentation

The retrieved data from shortlisted articles (see Table 1) have been presented in a tabular format (see Table 2), where the identified descriptive sensory terms were grouped into 5 categories: (1) smell/odourant; (2), flavour; (3) taste; (4) texture; and (5) visual. The presented words were accompanied by the reference letter from which they were initially extracted. Further information, like the word definition, has also been included for those that did possess one. The findings were then analysed and are discussed in the following sections.

## 3. Results

The reviewers have screened 827 articles, of which 12 were considered for data extraction. The majority of the screened studies did not meet the inclusion criteria (please see Appendix A). Interestingly, some peer-reviewed studies have not reported on the nature of the sensory ques used for the sensory evaluation. 

Out of the 12 articles, 205 descriptive sensory terms were identified, which were grouped into five categories: smell/odour, flavour, taste, texture, and visual. The table below shows the articles included for data extraction.

## 4. Applied Descriptive Sensory Terms for Peas

The total number of identified descriptive sensory terms was 205; this includes sensory properties identified through panel testing and volatile compounds identified instrumentally that have been assigned appropriate descriptive terms. This accounts for the five sensory attributes retrieved through the review: (1) smell/odourant—55 descriptive terms; (2) flavor—104 descriptive terms; (3) taste—21 descriptive terms; (4) texture—17 descriptive terms; and (5) visual—8 descriptive terms. The vast majority of the identified descriptive sensory terms for peas (*Pisum sativum* L.) differed between each other. The sensory attributes, most of the time, were developed by the panellists themselves during panel training and for products at different stages from across the value chain, hence the large number of different attributes being noted. Some of the selected studies have reported on an even greater number of descriptive sensory terms; however, those were shortlisted to the most common ones found amongst the panellists and through instrumental detection, thus leading to the total number of 205 of confirmed and reported attributes. Interestingly, most of the selected studies have defined their sensory attributes to an extent, but there were some misconceptions between the gustatory and olfactory senses. For example, the authors of study [E] have categorised odour and taste as different attributes in the initial stages of the study, but the results were all reported as flavour. Flavour is, in fact, the combination of smell and taste [24,25,39]; however, it is unclear how the panellists have distinguished the two different senses’ results through the gustatory testing of the prepared pea samples, especially as no indication of initial olfactory testing was reported, and only the “tasting” method was applied. In such a case, the identification of various volatile compounds could have been viewed as a more appropriate methodological approach.

## 5. Discussion

Despite the vast range of identified applied descriptive sensory terms, some of the attributes were repeated multiple times but classified under different senses. For example, some attributes have been used in three contexts, these were flavour, smell/odour, and taste, and “beany” has been linked to the volatile compound *3-Isobutyl-2-methoxypyrazine* (see B, E, G, F, I, J, and K). Furthermore, the “green” characteristic has been used as a sensory cue to describe the smell/odourant, flavour, and taste attributes of peas and has been accompanied by the specific volatile compound *hexanol* (see B, D, E, F, G, I, J, and K and Table 2). Most of these interconnections were found in between different studies, and very rarely in one study (see “earthy” for smell/odourant and taste for study H). Moreover, some of the applied descriptive sensory terms were modified to adjectival phrases like “earthy”, which has been modified to “earthy aroma” [B, E, H] or “bitter” -> “bitter taste” -> “bitter aftertaste” (see “flavour” for D, E, F). These linguistic constructs have been frequently used across the selected literature, thus increasing the number of identified records. Also, some very similar descriptive terms have been used interchangeably, for example, “cocoa” and “chocolate” [B] or “grassy” and “hay-like”, linked to the volatile compound *hexanal* (see “flavour” for A and F, respectively). Based on these examples, it can be argued that there are some major differences between these sensory attributes; however, it is worth mentioning that those distinctions were only noted between different studies and never within the same study, which could mean that the various sets of panelists have correctly identified the sensory attributes but then described them using parallel lexical terms [26]. All of this has led to the creation of a more diverse list of identified descriptive sensory terms for peas. Articles C, E, G, I, J, and K have included some definitions of their sensor constructs, making them less suspectable to human-based biases during sensory evaluation, some of which have been based on identified volatile compounds. 

### 5.1. Possible Implications of Species and Varietal Names

The descriptive sensory terms identified through this review were used to describe the attributes of peas, or what is taxonomically identified as *Pisum sativum* L. This rule has been avoided by one single study [B]; as mentioned previously, the Testa sample was described as “peas”, but no indication of species name was given; nevertheless, the study was considered as appropriate for inclusion. However, in one case, where the descriptive sensory terms were found as representative of *Pisum sativum* L., there were multiple results identified through the initial screening processes which were found to be of broad nature, where colloquial words like “legume”, “pea”, “bean”, and “pulse” were being manifested, instead of the actual species name. Something that is broadly known as a “bean” may sometimes refer to any species from the Fabaceae, or else specific species such as the “common bean” (*Phaseolus vulgaris* L.) or fava bean (*Vicia faba* L.) [2,4,40,41]. Another study has created sensory wheels for pulses and pulse-derived products but has not distinguished the differences between pulse species or types [42]. Simultaneously, the word “peas” is frequently used to describe pea-like vegetables, indicating the green colour of the round shaped seeds, green pea pods, and cultivation requirements, which leads to a fair conclusion that those belong to the same group or species of edible plants. The word “pea” is used interchangeably to describe actual peas (*Pisum sativum* L.) as well as completely separate species like pigeon peas (*Cajanus cajan* L.) and many others [40,43]. It is therefore necessary to state the correct species name [41].

The descriptive sensory terms for plants that seem similar in nature through a colloquial lens could prove to be significantly different [44]. Some of the broader sensory attributes like “beany” and “earthy” and associated volatile compounds would likely appear also in several other species from the Fabaceae family; however, the more specific attributes like “popcorn-like” or “nutty” (associated with volatile compound *2,3-Dimethylpyrazine*) could only be present for more limited numbers of taxa. Keeping this in mind, it can also be argued that varietal differences within one species could have a further impact of the sensory profile of these crops. Different varieties could show distinct attributes, such as the use of landraces of commercial cultivars (studies H and L) or subspecies, like in the case of study J, which adds to the pool of possible sensory attributes for a single type of a natural product. Therefore, it is extremely important to note what the tested species were, and especially what the varieties were, where and when they were cultivated, and for what purpose [45]. Through this review, based on the methodologies stated within the included studies, it can be assumed that around 43 different varieties of peas were tested, which can be accepted as a representative sample of the species of *Pisum sativum* L. Studies A, B, C, F, G, H, J, and K have tested the sensory attributes of 1 to 3 different varieties/cultivars of peas. Studies D, E, I, and L have used between 5 and 12 pea varieties. Moreover, the selected literature lists at least eight countries (China, Denmark, Germany, Italy, Poland, South Africa, Sweden, and the UK) in which the samples were initially cultivated, which further strengthens the results’ diversity, especially as some of the identified descriptive sensory terms were of a similar nature despite distinct sample collection and cultivation environments. 

Apart from the species and varietal variables for the pea sensory analyses, sample preparation and testing methodologies have themselves inflicted biases for the results [24,46]. For example, pea samples made using pea protein isolates were served in liquid forms as water dilutions (see study B and D), whereas some (e.g., study coded A) were tested as raw peas or after minor processing, for example, steaming, roasting, boiling, or freezing. The nature of the sample determines the specific sensory attributes that are present within and could promote selection of their favourable attributes over others [23,24,46]. The impact of these processing methods on the sensory characteristics is often significant and could therefore be the reason for such a diverse list of descriptive sensory cues identified through this systematic literature review [47,48]. As such, peas that have been blanched, frozen, and boiled for serving would have undergone extensive physiochemical changes, impacting the type of starches present as well as the abundance, intensity, and type of volatile compounds and other chemicals responsible for visual characteristics, including the degradation of *chlorophyll*, oxidation, or enzymatic browning. On the other hand, such treatments can also preserve the intensity of some sensory characteristics like “colour”, for example, by slowing down the enzymatic browning through exposure to heat or manipulation of the pH. Heat treatment often decreases the abundance of volatile compounds that are associated with specific sensory characteristics. In peas, these could be “fresh” or “green” flavours, which are linked to volatile compounds like *hexanal* and the promotion of other compounds like *alkaloids*, which are linked to other flavours, including “bitter”, creating the diverse range of sensory characteristics throughout the product’s life cycle (see Table 2). Furthermore, significant changes to texture often occur after the blast-freezing of peas, and the types of volatile compounds present within those and fresh peas are often different and detectable instrumentally, but also through tasting, increasing the diversity within the lexicon [23,24]. 

Study J was based on fermented pea products for which the intensity of identified descriptive sensory terms changed over time; however, the terms themselves have remained the same throughout the 28-day-long experiment. Fermentation is responsible for the creation of new physiochemical properties in foods, including changes to existing volatile compounds, as well as the creation of completely new compounds. 

Most of the selected studies have developed their own descriptive sensory terms for the specific samples, making the results more bias-resistant [44]. Nevertheless, the risks persisted, as all sensory trials were liable to subjectivity imposed by the panellists as well as the researchers and data analysists. Using trained panellists instead of untrained panellists is a much more coherent method for sensory analyses, especially when working on unique or new products and training to establish descriptive sensory lexicons and product-descriptive sensory profiling. Untrained panellists could prove very useful for comparison or hedonic tests but would be less likely to distinguish a larger number of specific sensory attributes [23,24,45,46]. Furthermore, a wider application of instrumental methods, together with the panel-based methods, is considered the most appropriate approach in such evaluations as it ensures maximum objectivity but with considerations towards the subjective elements of human sensory palates. In relation to products such as peas, it is recommended that more attention is dedicated towards aspects such as species or cultivars/varieties, cultivation methods and locations, processing, and storage as each one of these can impose significant changes on the sensory profile of the product, thus leading to different outcomes at various steps along the products’ value chains [24].

Eight out of the twelve articles [A, B, C, E, G, I, K, and L] have reported on using trained panellists, with numbers between nine and twelve, giving a total number of eight-five trained panellists. Out of the total number, the majority were females. The age gap spans between 23 and 50; however, some studies [C E, J, K, and L] have not stated this information. One study [D] has not included any information on the panellists’ training or demographics. Study F was a systematic literature review; thus, no panellists were included, and studies H and J consisted of 26 and 10 untrained panellists, respectively. The diversity of the various panellists could be seen as an advantage and to add validity to the identified lists, making this review’s results more versatile. 

### 5.2. Results Synthesis and Limitations

An attempt to synthesise the results was carried out via the removal of synonyms and synonym-like constructs from the generated list of descriptive sensory terms for the sensory evaluation of peas. For the purpose of the data synthesis process, “synonym” has been defined as a word that represents an exact or similar sensory cue to the original word, for example: *bean* and *pulse* [27]. Furthermore, constructs like *grassy* and *grass-like* or *beany* and *bean-like*, where prepositions and conjunctions were used in the form of the word “like” and others, have also been removed from the final list of descriptive sensory terms.

The gustatory and olfactory results combined were narrowed to 189 descriptive sensory terms (see Table A1). The possible implications of this synthesis of the descriptive terms could result in a more efficient sensory evaluation of peas as less constructs would have to be analysed by the panel. On the contrary, the smaller number of descriptive terms could increase the subjectivity and decrease the trustworthiness of the results, making the hypothetical study less valid; therefore, it should be accompanied by other analyses, including the detection of existing volatile compounds [45]. However, over half of the selected studies only included around 12 descriptive sensory terms for the sensory evaluation, which could be an argument in favour of narrowing down the scope of the possible synthesis of the identified descriptive sensory terms. No information has been found to confirm the ideal number of descriptive sensory terms for legumes and peas specifically, but multiple sources suggested maximising the number of descriptive sensory terms [23,24,45,46]. Good sensory evaluation practices dictate that “new” products would be undertaken for a full evaluation in search of all the present descriptive sensory terms and for instrumental detection of the key volatile compounds and other characteristics throughout the product’s life cycle [13]. This methodology has been followed by some of the selected studies, where the panel had to develop a list of descriptive sensory terms and then shortlist it to a “manageable list”. On that note, it would be advisable for any future sensory evaluations of *Pisum sativum* L. to be initially carried out with the full list of the identified descriptive sensory terms through this review, selecting only the most appropriate attributes for the final evaluation as those could differ between the specific varieties of peas or other pea-derived products under investigation. 

## 6. Conclusions

Although legumes and peas are an important group of crops for the future of food and nutrition security, consumer acceptance through sensory preference is an important obstacle on the path to sustainability. Understanding the sensory attributes of peas on the inter-varietal spectrum could help to promote some of the neglected varieties, leading to an increase in agrobiodiversity, but this could also pose some positive implications for consumers from gastronomical and nutritional perspectives. 

This review shows that there is a wide list of different descriptive sensory terms and constructs a comprehensive descriptive sensory lexicon for peas (*Pisum sativum* L.) detected through the sensory evaluation of panelists and instruments (in the papers reviewed). The 12 shortlisted articles provided 205 descriptive sensory terms for peas, grouped into five senses, i.e., smell/odour, flavour, taste, texture, and visual, with some minor definitions and linkages to volatile compounds. After the analysis, it has been concluded that with the removal of duplicates, synonyms, and synonym-like constructs, the list could be limited to 189 descriptive sensory terms. On the other hand, it could be argued that sensory evaluation of products should consist of the maximum number of relevant sensory attributes, especially for the initial/training sessions, and that only the most accurate descriptive terms should be used for the final sensory evaluation. Nevertheless, using the formed descriptive sensory lexicon for the evaluation of peas is likely to increase the standardisation of the process in the future.

The recommendations for future systematic reviews in this area should diverge from the common commercial varieties of peas and focus more on the non-commercial, neglected, and underutilised relatives, which could be of great utility to researchers and businesses alike. These are likely to be rich in interesting sensory characteristics, which can be exploited by the market, the food industry, and wider food systems. Furthermore, the authors recommend further systematic literature reviews focusing on processed pea products that have not been shortlisted in this review.

## Figures and Tables

**Figure 1 foods-13-02290-f001:**
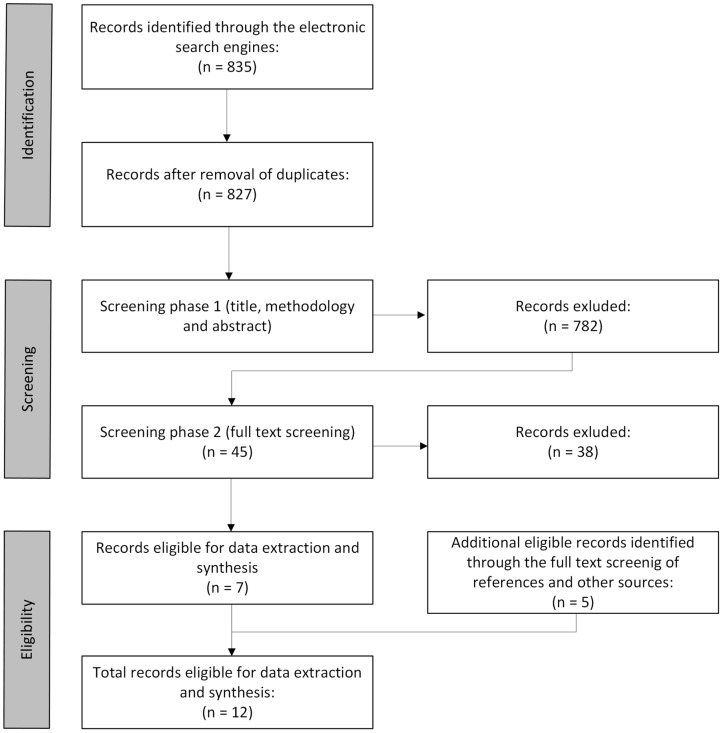
PRISMA flow chart. The chart represents the evidence search and shortlisting process. “n” = number of records of evidence.

**Table 1 foods-13-02290-t001:** Articles included for the data extraction. The articles have been listed and are referred to through the text using alphabetical letters (A–L).

Alphabetical Order:	Full Reference:
A [27]	Bi S, Lao F, Pan X, Shen Q, Liu Y, Wu J., 2022. *Flavor formation and regulation of peas (Pisum sativum L.) seed milk* via *enzyme activity inhibition and off-flavor compounds control release*. *Food chemistry*. 380(Journal Article):132203–132203.
B [28]	Bi S, Xu X, Luo D, Lao F, Pang X, Shen Q, Hu X, Wu J., 2020. *Characterization of key aroma compounds in raw and roasted peas (Pisum sativum L.) by application of instrumental and sensory techniques*. *Journal of agricultural and food chemistry*. 68(9):2718–27.
C [29]	Edelenbos M, Thybo A, Errichsen L, Wienberg L, Andersen L., 2001. *Relevant measurements of green pea texture. Journal of food quality*. 24(2):91–110.
D [30]	García Arteaga V, Kraus S, Schott M, Muranyi I, Schweiggert-Weisz U, Eisner P., 2021. *Screening of Twelve Pea (Pisum sativum L.) Cultivars and Their Isolates Focusing on the Protein Characterization, Functionality, and Sensory Profiles. Foods*. 10(4):758.
E [31]	Nleya KM, Minnaar A, de Kock HL., 2014. *Relating physico-chemical properties of frozen green peas (Pisum sativum L.) to sensory quality: Sensory and physico-chemical properties of frozen green peas*. *Journal of the science of food and agriculture*. 94(5):857–65.
F [32]	Roland WSU, Pouvreau LAM, Curran J, Velde van de Fred, Kok de Peter MT., 2017. *Flavor Aspects of Pulse Ingredients*. *Cereal chemistry*. 94(1):58–65.
G [33]	Troszyńska A, Szymkiewicz A, Wołejszo A., 2007. *The effects of germination on the sensory quality and immunoreactive properties of pea (Pisum sativum L.) and soybean (Glycine max)*. *Journal of food quality*. 30(6):1083–100.
H [34]	Westling M, Leino MW, Nilsen A, Wennström S, Öström Å., 2019. *Crop and livestock diversity cultivating gastronomic potential, illustrated by sensory profiles of landraces. Journal of food science*. 84(5):1162–9.
I [35]	Zhogoleva A, Alas M, Rosenvald S. *Characterization of odor-active compounds of various pea preparations by GC-MS, GC-O, and their correlation with sensory attributes*. *Future Foods*. 2023 Dec 1;8:100243.
J [36]	Demarinis C, Verni M, Pinto L, Rizzello CG, Baruzzi F. *Use of selected lactic acid bacteria for the fermentation of legume-based water extracts. Foods*. 2022 Oct 25;11(21):3346.
K [37]	Arteaga VG, Leffler S, Muranyi I, Eisner P, Schweiggert-Weisz U. *Sensory profile, functional properties and molecular weight distribution of fermented pea protein isolate. Current research in food science*. 2021 Jan 1;4:1–0.
L [38]	Westling M, Leino MW, Wennström S, Öström Å. *Sensory variation of landrace peas (Pisum sativum L.): Impacts of variety, location, and harvest year*. *Food Science & Nutrition*. 2024.

**Table 2 foods-13-02290-t002:** Findings representing all the identified descriptive sensory terms for peas. The first column describes the identified linguistic terms used for the descriptive evaluation of peas. The second column shows the reference codes corresponding to the evidence sources presented in Table 1. The third column shows any identified definitions for the linguistic terms. Some of the definitions also consist of identified volatile compounds and other chemical compounds, which have been presented in *italics*.

Linguistic Terms:	Reference Codes:	Definitions Identified:
Smell/Odourant:		
Almond	(Ref no B)	
Beany	(Ref no G)	“Odour characteristic for boiled dry legume seeds”
Beany, green-peppers	(Ref no I)	“*3-Isobutyl-2-methoxypyrazine* (IBMP)”
Burnt sugar	(Ref no B)	
Buttery	(Ref no K and L)	Ref K: “*2,3-butanedione*”
Cabbage-like	(Ref no B)	
Cacao	(Ref no L)	
Cacao-like	(Ref no B)	
Cereal	(Ref no L)	
Cheesy	(Ref no K)	“*3-methylbutanoic acid*”
Chocolate-scented	(Ref no B)	
Citrus, fresh	(Ref no I)	“*decanal*”
Cucumber, green	(Ref no I)	“*(E,Z)−2,6-Nonadienal*”
Cucumber-like	(Ref no B)	
Earthy	(Ref no H, K and L)	Ref K: “*geosmin*”
Fatty	(Ref no I)	“*(E,E)−2,4-Decadienal*”
Fecal	(Ref no K)	
Fermented	(Ref no K and L)	
Floral	(Ref no B, J and L)	
Floral, soapy	(Ref no I)	“*2-Decanone*”
Floury	(Ref no K)	
Fresh-cut grass	(Ref no J)	
Fruity	(Ref no J)	
Gammon-like	(Ref no B)	
Garlicy	(Ref no B)	
Grassy	(Ref no G)	“Odour associated with freshly cut grass”
Grassy, green-apple	(Ref no I)	“*hexanal*”
Greasy	(Ref no K)	“*2-nonenal*”
Green	(Ref no B and K)	Ref K: “*haxanal*”
Green flavour/Vegetal	(Ref no J)	
Green peas	(Ref no J)	
Green-peas, peapods	(Ref no I)	“*3-Isopropyl-2-methoxypyrazine* (IPMP)”
Herbaceous	(Ref no B)	
Honey-like, sweet	(Ref no I)	“*Phenylacetaldehyde*”
Iodoform	(Ref no B)	
Leguminous-plant	(Ref no J)	
Malty	(Ref no B and I)	Ref I: “*3-Methylbutanal*”
Mushroom-like	(Ref no B)	
Nutty	(Ref no B, I and L)	Ref I: “*2,3-Dimethylpyrazine*”
Oatmeal	(Ref no K)	
Odour intensity	(Ref no H)	
Phenolic	(Ref no B)	
Popcorn-like	(Ref no B)	
Potato-like	(Ref no B)	
Rancid	(Ref no G)	“Odour connected with aged oil”
Roasted	(Ref no B, K and L)	Ref K: “*furaneol/acetylpyridine*”
Roasted nut	(Ref no B)	
Root fruit	(Ref no L)	
Rubber	(Ref no B)	
Sickly	(Ref no B)	
Smoky	(Ref no B)	
Soapy	(Ref no B)	
Spicy	(Ref no K)	“*Sotolone*”
Sulphur/sulfur	(Ref no B and I)	Ref I: “*methanethiol*”
Vegetable-like	(Ref no B)	
Flavour:		
Acetone aroma	(Ref no E)	“Aromatic characteristic of *ketones*, specifically *acetone*”
Acorn	(Ref no L)	
Acrid	(Ref no F)	
Almond	(Ref no L)	
Aroma intensity	(Ref no E)	“The strength of odour that is released from the peas upon taking the first few sniffs”
Asparagus	(Ref no L)	
Astringent	(Ref no D and F)	
Beans-like	(Ref no B)	
Beany	(Ref no F)	
Beany aroma	(Ref no E)	“Aromatic characteristic of *leguminous* seeds”
Bitter	(Ref no D, F and L)	
Bitter aftertaste	(Ref no E)	“Intensity of bitter taste that lingers after swallowing”
Bitter taste	(Ref no E)	“Taste on tongue stimulated by *caffeine*, *quinine* and certain other *alkaloids*”
Brazil nut	(Ref no L)	
Brothy	(Ref no F)	
Buttery	(Ref no L)	
Cacao	(Ref no L)	
Cereal	(Ref no L)	
Chestnut	(Ref no L)	
Chocolate	(Ref no L)	
Citrus	(Ref no L)	
Compact	(Ref no L)	
Compost	(Ref no L)	
Corn	(Ref no L)	
Crunchy	(Ref no L)	
Dairy protein	(Ref no L)	
Dry	(Ref no L)	
Earthy	(Ref no A, D, F and L)	
Earthy aroma	(Ref no E)	“Aromatic characteristic of damp soil, wet foliage or undercooked potato”
Fatty	(Ref no A, B, D, F and L)	
Fermented	(Ref no L)	
Flavour intensity	(Ref no E)	“Strength of flavour concentration released in mouth when pea sample is chewed”
Floral	(Ref no L)	
Fresh flavour	(Ref no E)	“Flavour associated with fresh green peas, free from any unfavourable/stale odours”
Fruity	(Ref no L)	
Fruity flavour	(Ref no E)	“Flavour associated with fully ripened fruit characteristic of *aldehydes* and *ketones*”
Grainy	(Ref no L)	
Grass-like	(Ref no B)	
Grassy	(Ref no A and L)	
Green	(Ref no D and F)	
Green apple	(Ref no L)	
Green aroma	(Ref no E)	“Aromatic associated with freshly cut green vegetables”
Green Pea	(Ref no L)	
Hay	(Ref no L)	
Hay-like	(Ref no F)	
Hazelnut	(Ref no L)	
Herby	(Ref no L)	
Iron	(Ref no L)	
Juicy	(Ref no L)	
Lard	(Ref no L)	
Leafy	(Ref no F)	
Licorice	(Ref no L)	
Malty	(Ref no D)	
Meaty	(Ref no L)	
Metallic	(Ref no B and D)	
Metallic	(Ref no L)	
Milk chocolate	(Ref no L)	
Milk-like	(Ref no A)	
Mineral	(Ref no L)	
Mint	(Ref no L)	
Mouth-coating	(Ref no D)	
Mushroom	(Ref no L)	
Musty	(Ref no L)	
Nutty	(Ref no B, D and L)	
Oats	(Ref no L)	
Pea	(Ref no F)	
Pea-like	(Ref no D)	
Pepper	(Ref no L)	
Pepper (pepper-corn)	(Ref no L)	
Popcorn	(Ref no L)	
Popcorn-like	(Ref no B)	
Porous	(Ref no L)	
Potato	(Ref no L)	
Potato-like	(Ref no B)	
Pungent	(Ref no F)	
Raw beans	(Ref no A)	
Roasted	(Ref no L)	
Roasty	(Ref no D)	
Root fruit	(Ref no L)	
Salt	(Ref no L)	
Salty	(Ref no D)	
Shiitake	(Ref no L)	
Silage	(Ref no L)	
Smoky	(Ref no B and L)	
Soft	(Ref no L)	
Sour	(Ref no L)	
Sour odour	(Ref no L)	
Spicy	(Ref no L)	
Stable	(Ref no L)	
Starchy flavour	(Ref no E)	“Flavour associated with tubers particularly boiled potato”
Sulphur/sulfur	(Ref no L)	
Sunflower seed	(Ref no L)	
Sweet	(Ref no D and L)	
Sweet aroma	(Ref no E)	“Aromatic associated with high sugar content vegetables”
Sweet taste	(Ref no E)	“Taste on tongue stimulated by sugars and high potency sweeteners”
Tender	(Ref no L)	
Tough	(Ref no L)	
Umami	(Ref no L)	
Vegan butter	(Ref no L)	
Vegetable	(Ref no L)	
Walnut	(Ref no L)	
Woody	(Ref no L)	
Yeasty	(Ref no L)	
Yellow apple	(Ref no L)	
Taste:		
Acid	(Ref no K)	
Astringent	(Ref no G and I)	Ref G: “Taste illustrated by unripe banana (reference sample: *tannic acid* 0.2%)” Ref I: “Chemical feeling factor characterized by a drying or puckering of the oral tissues”
Beany	(Ref no G)	“Taste typical for boiled legume seeds”
Bitter	(Ref no G, H, I, K and L)	Ref H: “Basic taste (reference sample: *caffeine* in water 0.5%)” Ref I: “Basic taste elicited by various compounds including *caffeine* and *quinine*”
Buttery	(Ref no L)	
Cacao	(Ref no L)	
Corn	(Ref no L)	
Earthy	(Ref no H)	
Fresh	(Ref no G)	“Taste of fresh vegetables (*versus* processed products)”
Grassy	(Ref no G)	“Specific note characteristic for green pea-pod without seeds”
Green	(Ref no G)	“Taste characteristic for fresh green pea seeds”
Herby	(Ref no L)	
Nutty	(Ref no H and L)	
Pungent	(Ref no G)	“Taste associated with fresh *cruciferous* vegetables (reference sample: slice of radish)”
Rancid	(Ref no F)	“Taste characteristic for aged fat oil (reference sample: *iso-butyric acid* 0.1%)”
Salt	(Ref no L)	
Sour	(Ref no L)	
Spicy	(Ref no L)	
Sweet	(Ref no H and L)	
Taste intensity	(Ref no H)	
Umami	(Ref no L)	
Texture:		
Chewiness	(Ref no E)	“Amount of work required to masticate a pea sample with molars”
Chewing resistance	(Ref no H)	
Cotyledon hardness	(Ref no C)	“Force required biting completely through *cotyledons* from three pea seeds at first bite when placed between molars”
Crispiness	(Ref no C)	“Place sample between molars, bite through and evaluate for level of higher pitched noise at first bite”
Crunchiness	(Ref no E)	“Pitch of sound produced when chewing peas”
Dry	(Ref no H)	
Fibrousness	(Ref no G)	“Feeling of ’fibrousness’ perceived while chewing the sample 10 times”
Hard	(Ref no H)	
Juiciness	(Ref no C and G)	Ref C: “Total amount of juice released on chewing” Ref G: “Feeling of juiciness perceived while chewing the sample 10 times”
Mealiness	(Ref no C and E)	Ref C: “*Mealiness* or *grittiness* is the starch-like sensation felt between the tongue and roof or sides of mouth when *cotyledons* from 3 peas are chewed 2–3 times and the residue pressed against roof of the mouth” Ref E: “Extent of *granularity* in texture experienced when chewing peas”
Mealy	(Ref no H)	
Moistness	(Ref no E)	“The amount of juice released from peas upon chewing a spoon full of peas”
Residue remaining after swallowing	(Ref no E)	“The amount of pea pieces that remain in mouth after chewing and swallowing”
Seed hardness	(Ref no C)	“Force required biting completely through peas at first bite when placed between molars”
Tenderness	(Ref no E)	“Ease with which peas are masticated in the mouth”
Testa toughness	(Ref no C)	“Testa (skin) toughness or skin chewiness is the total amount of work necessary to chew 3 Testas to a state ready for swallowing. Tender skin is < 10 chews and tough skin > 35 chews”
Uniformity in texture	(Ref no E)	“Estimated level of homogeneity in texture in a spoonful of peas”
Visual:		
Exterior seed surface texture	(Ref no E)	“Degree of shrivelling of outer skin surface”
Green colour intensity	(Ref no E)	“Level of greenness of outer surface of peas”
Green colour uniformity	(Ref no E)	“Estimated level of homogeneity in colour of pea seeds”
Greenish-yellow colour	(Ref no J)	
Overall seed shape uniformity	(Ref no E)	“Estimated level of homogeneity in shape of pea seeds”
Seed shape	(Ref no E)	“Characteristic surface outline/fullness of pea seeds”
Seed size	(Ref no E)	“Physical dimensions of pea seeds”
Seed size uniformity	(Ref no E)	“Estimated level of homogeneity in size of pea seeds”

## Data Availability

Data sharing is not applicable to this article as no datasets were generated or analysed during the current study.

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
