# Peer review of "A Lexicon of Descriptive Sensory Terms for Peas (Pisum sativum L.): A Systematic Review"

_foods, 2024, doi:10.3390/foods13142290_

Round 1

Reviewer 1 Report

Comments and Suggestions for Authors

Manuscript 3079637

Journal Foods

Title A Lexicon of Descriptive Sensory Terms for Peas (Pisum sativum L.): A Systematic Review

The review entitled “A Lexicon of Descriptive Sensory Terms for Peas (Pisum sativum L.): A Systematic Review” describes the terms associated with sensory properties of peas. Eight articles were selected for the analysis of descriptive sensory terms of smell/odour, flavour, taste, texture and visual quality. Eight articles for a systematic review are too few for the acceptance of publication in Foods. Please enrich the review with the descriptive terms of processed peas (e.g., pea preparations or fermented pea products). Moreover, a link between sensory properties and volatile compounds of peas should be highlighted throughout the manuscript. Please follow the comments in the file.

Comments on the Quality of English Language

Author Response

Dear reviewer, please see the file attached for our comments. 

Reviewer 2 Report

Comments and Suggestions for Authors

Dear authors, the revised manuscript is interesting. It is recommended to attend to the following:

Line 3: insert text space…: A

Line 4: use superscript text formatting in affiliation numbers

Line 30,31: remove the number used in each of the keywords

Line 30,31: It is recommended that keywords should be written using no more than three words in each of them

Line 41: remove text space… [1,2].

Line 41: It is recommended that you use the abbreviated scientific name from the second time it appears in the text and throughout the document.

Line 43,44: check correct text formatting

Line 44: use [4–6] instead of [4-6]

Line 51: remove text space… [10,11]

Line 53: remove text space… [6,8,10,11]

Line 55: remove text space… [11,12]

Line 81: remove text space… [10,18,19]

Line 83: remove text space… [7,20,21]

Line 87: remove text space… [20,23]

Line 95: remove text space… [24,25]

Note: In Tables 1 and 2, check that the font type and size are those requested in the journal's author guide. Homogenize reference format, complete missing information in references, and use italic text format for scientific names.

Line 178: remove text space… [24,25,27]

Line 214: remove text space… [2,4,28,29]

Line 214: According to the author guide, it is necessary not to mention names or surnames of authors in the text, it is only suggested to put the corresponding reference number in square brackets.

Line 221: It is recommended that you use the abbreviated scientific name from the second time it appears in the text and throughout the document.

Line 221: remove text space… [28,31]

Line 248: remove text space… [23,24,34]

Line 248: remove text space… [23,24,33,34]

Line 266: remove :

Line 284: remove text space… [23,24,33,34]

Line 329: check that the font type and size are those requested in the journal's author guide

Line 359: check in the author guide the correct text format for this section

Author Response

(The authors gave the same response as above.)

Reviewer 3 Report

Comments and Suggestions for Authors

This work aims to identify descriptive sensory terms for peas from published articles. The results might be helpful for the descriptive sensory evaluation of peas. There are several issues that need to be addressed.

1.        The keywords are too limited. I suggest they be changed to ‘pea; Pisum sativum L.; sensory characteristic; descriptive sensory term’.

2.        When conducting the literature search, why not include the database of Web of Science? As is known, Web of Science includes more databases such as Springerlink, Wiley, etc.

3.        Line 110: should ‘sensor* analys*’ be ‘sensory* analysis*’?

4.        Only 8 articles were screened for data extraction. This is a limitation of this research. It seems too little to reach a reliable conclusion. Please give some discussions on this limitation.

5.        Appendix 1: add the categories of the terms.

Author Response

Dear reviewer,  please the file attached for our comments. 

Round 2

Reviewer 1 Report

Comments and Suggestions for Authors

Auhtors revised the original manuscript according to reviewer's comments. However, some changes are still necessary prior the acceptance for publication in Foods. Please follow the comments below:

L173-193 Please include the volatile compounds associated with particular sensory notes detected in these studies (e.g., hexanal associated with the green notes). The link between volatile compounds and sensory characteristics is lacking in the manuscript. Please revise the whole manuscript according to this aspect.

L265-268 Please add more details on these aspects. How processing conditions affect the sensory characteristics? Please be more specific

L403-473 Include the references of Table 1 here, in the reference list. Thanks

Comments on the Quality of English Language

Moderate changes are necessary

Author Response

Hi there,

Thank you again for the second round of comments. 

Please see the document attached with our comments.

Best

Reviewer 3 Report

Comments and Suggestions for Authors

The manuscript had been improved according to the comments. I think it could be accepted for publication with some editorial changes.

Author Response

Thank you for the time and the feedback provided. Much appreciated.